# MAD-FC: A fold change visualization with readability, proportionality, and symmetry

**Bruce A. Corliss**[1,2]*, **Yaotian Wang**[3], **Francis P. Driscoll**[1], **Heman Shakeri**[1], **Philip E. Bourne**[1,2]

**1** School of Data Science, University of Virginia, Charlottesville, Virginia, United States of America, **2** Department of Biomedical Engineering, University of Virginia, Charlottesville, Virginia, United States of America, **3** Department of Statistics, University of Pittsburgh, Pittsburgh, Pennsylvania, United States of America

* bac7wj@virginia.edu

**Data Availability Statement:** Code used to generate all data and figures is written in R and available at: https://github.com/bacorliss/mirrored_axis_distortion.

## Abstract

We propose a fold change transform that demonstrates a combination of visualization properties exhibited by log and linear plots of fold change. A fold change visualization should ideally exhibit: (1) readability, where fold change values are recoverable from datapoint position; (2) proportionality, where fold change values of the same direction are proportionally distant from the point of no change; (3) symmetry, where positive and negative fold changes of the same magnitude are equidistant to the point of no change; and (4) high dynamic range, where datapoint values are distinguishable across orders of magnitude within a fixed plot area and pixel resolution. A linear visualization has readability and partial proportionality but lacks high dynamic range and symmetry (because negative direction fold changes are bound between [0, 1] while positive are between $(1, \infty)$). Log plots of fold change have partial readability, high dynamic range, and symmetry, but lack proportionality because of the log transform. We outline a new transform, named mirrored axis distortion of fold change (MAD-FC), that extends a linear visualization of fold change data to exhibit readability, proportionality, and symmetry (but still has the limited dynamic range of linear plots). We illustrate the use of MAD-FC with biomedical data using various fold change plots. We argue that MAD plots may be a more useful visualization than log or linear plots for applications that do not require a high dynamic range (less than 8 units in log2 space).

## Introduction

Bioinformatics research often requires analyzing datasets that are expressed in units of fold change [1]. This measurement type represents the ratio of the sample mean of an experiment group divided by a control group. Fold change data is visualized to summarize the spread of the dataset and identify interesting datapoints- often those with the largest magnitude in either direction of change. Typically, scientists visualize fold change using a log [2] or a linear transform, the latter of which simply presents raw fold change values [3]. Each transform has a unique set of properties that may be advantageous or disadvantageous depending on the particular purpose of the visualization. We propose a transform, mirrored axis-distortion of fold

**Funding:** BAC This work was funded by PEB's endowment for the School of Data Science, University of Virginia. The funder had no role in study design, data collection and analysis, decision to publish, or preparation of the manuscript.

**Competing interests:** The authors have declared that no competing interests exist.

change (MAD-FC), and demonstrate its use with visualizing real research data. Our proposed fold change transform may enhance the understanding and interpretation of fold change data in scientific research because it combines some of the more useful properties of linear and log transforms.

To assist with our discussion of fold change visualization properties, we introduce the term *point of no change*, which is the value in a fold change transform space that denotes no change and separates negative fold changes from positive fold changes. For a log transform of fold change measurements, the point of no change is zero; for a linear transform, the point of no change is one. We also define a linear encoding for fold change, *fold change units*, that represents the number of fold changes from the point of no change. With this encoding, the raw fold changes of (2, ½, 3, 1/3) are mapped to (1, -1, 2, -2) fold change units, respectively. Fold change units, $f_U$, can be mathematically defined as:

$$f_U(x) = \begin{cases} x - 1 & x \geq 1 \\ 1 - \dfrac{1}{x} & x \in (0, 1), \\ undefined & otherwise \end{cases} \quad (1)$$

where $x$ is the raw fold change measurement. With these terms defined, we can now propose the properties of a useful fold change visualization and then evaluate various types of plots (summary in Fig 1A):

1. **Readability**: a visualization exhibits readability if it allows the observer to easily recover the original fold change values from the datapoints within the visualization. To fulfill this condition, there must be a clear and direct mapping between the values of the visualized datapoints and their spatial locations. This property can be split into two steps: (1) *extraction*, where a datapoint value is extracted from the plot coordinate system based on its relative position to the nearest axis tick labels, and (2) *conversion*, where the extracted value is converted to the underlying fold change value (if necessary). For example, for a plot using raw logarithmic axis tick labels (e.g. inner y-axis labels of Fig 1D), the log transform must be reversed on each datapoint extracted from the plot's coordinate system to obtain the original fold change measurement. A visualization that is readable facilitates rapid extraction and conversion of its datapoints with minimal effort from the observer.

2. **Proportionality**: a visualization exhibits proportionality if the fold change values are proportionally distant from the point of no change within the plot. This proportional relationship does not have to be identical for positive versus negative fold changes. Proportionality allows for direct comparisons between the magnitudes of positive fold changes (and separately for negative fold changes). This property holds true if there is an exact linear relationship between the transformed fold changes and corresponding fold change units for both positive fold changes and negative fold changes. Proportionality can be visually assessed by generating a test dataset of fold change datapoints and plotting the transformed value versus its fold change units. Within this plot, we connect a line between the largest positive fold change and the point of no change. If all positive fold change datapoints fall on the line, then positive fold changes are designated as proportional for the transform. The same procedure can be used to examine negative fold changes. If both fold change directions are proportional, then we designate the transform as having proportionality. Note that the slopes for these two lines are not required to be one (meaning they do not need a direct 1:1 correspondence to fold change units), and the slope does not have to be the same between directions (that is examined with the next property).

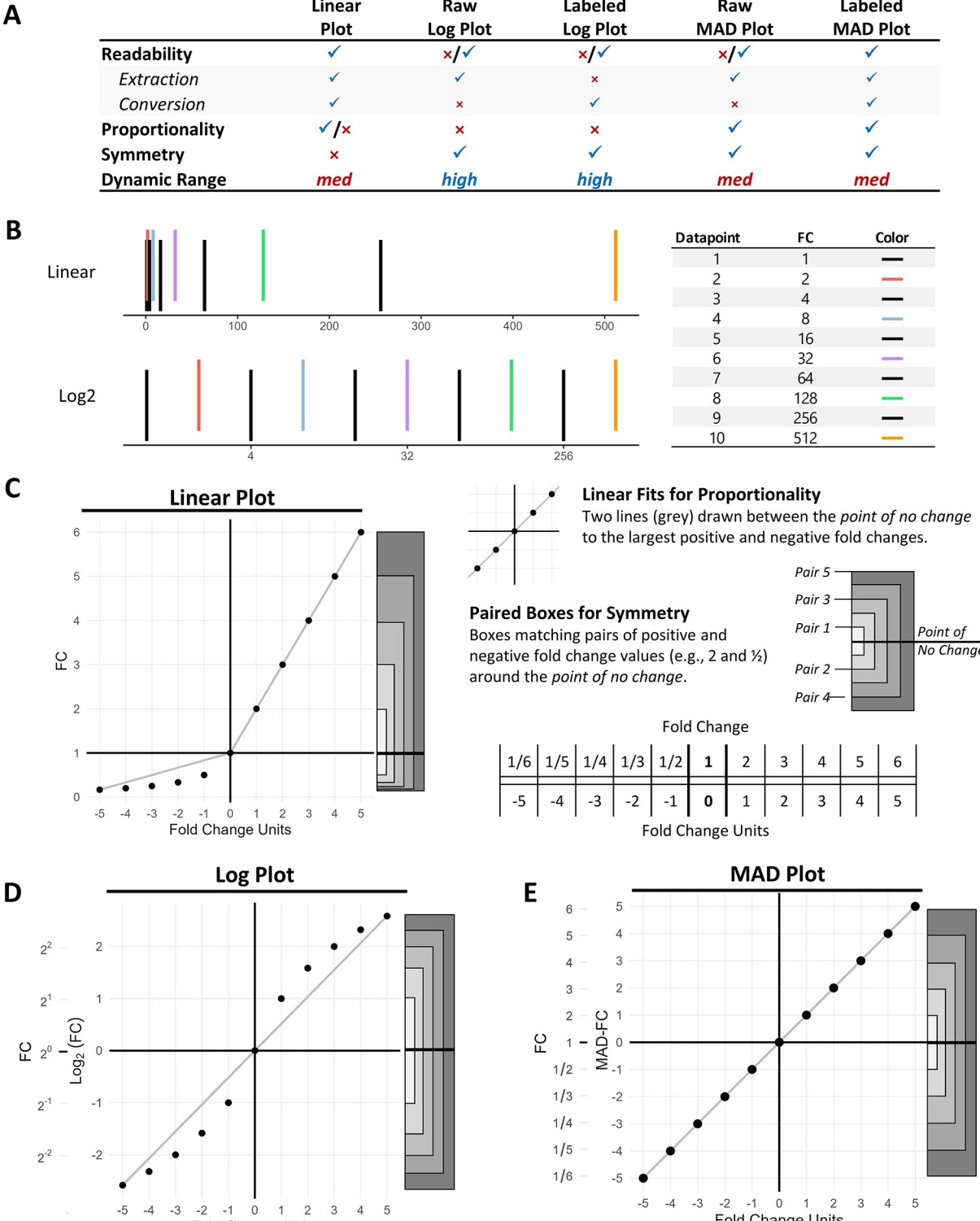

**Fig 1. Evaluating the visualization properties of linear, log, and MAD plots.** (**A**) Table summarizing visualization properties for each of the plot types. (**B**) Visualizing a dataset of positive fold changes with a linear (upper) and log2 (lower) scale to illustrate dynamic range (each datapoint is a staggered line, alternating between black and another color to aid with datapoint identification). (**C-E**) A dataset of fold changes ranging from 1/6 to 6 (equal to -5 to 5 in fold change units, see legend on right) is visualized with (**C**) linear, (**D**) log, and (**E**) MAD plots to illustrate their characteristics. Visualization properties of plot types are assessed with: readability based on the transform and the units of the axis tick labels, proportionality from linear fits between the point of no change and extreme datapoints (grey lines), symmetry from boxes that

match points of same magnitude (grey shaded boxes to right of each plot), and dynamic range based on whether datapoints are mapped to linear space (medium) or log space (high). (**D**-**E**) Inner y-axis tick labels are raw transform units; outer y-axis tick labels are back-transformed to original fold change units.

3. **Symmetry**: a fold change plot has symmetry if every datapoint would remain equidistant to the point of no change if its fold change direction were hypothetically reversed. This property helps scientists visually compare the magnitudes of positive and negative fold changes. Symmetry of a transform can be visually assessed by measuring the distance between synthetically generated pairs of fold changes of opposite direction with the same magnitude. For example, a fold change of 2 should be equidistant to the point of no change compared to a fold change of ½, a fold change of 3 should be equidistant compared to a fold change of 1/3, and so forth.

4. **Dynamic Range**: fold change measurements can span many orders of magnitude, and for some applications it is important to clearly distinguish the positions of points across this range. We illustrate this by visualizing a dataset of fold changes ranging from a value of 1 to $2^9$ in magnitude on a linear and log scale (Fig 1B). It is easy to perceive the distinct lines when the datapoints are visualized with a log2 transform, but the smallest fold changes are difficult to perceive on a linear scale. When data spans multiple magnitudes on a linear scale, large outlying data values overwhelm the axis spatial encoding, often leaving insufficient space to distinguish differences between small values (e.g. the crowding between small fold changes on the linear axis in Fig 1B). While the number of orders of magnitude that a linear scale can capture depends on the pixel resolution of the plot, image compression artifacts, plot area, visual acuity, and viewing distance, we estimate a plot with a linear scale can typically span $2^8$ units (approximately $10^{2.5}$ units in base 10). Based on this basic characterization, we designate a log transform as having high dynamic range, while a linear transform has medium dynamic range. It is important to note that plots with log transforms are not universally better at distinguishing closely positioned points when compared to linear- there are instances where two datapoints are located closer together in a log plot than a corresponding linear plot and therefore harder to distinguish.

It is important to note that while these properties are explained in the context of spatial encodings, they could potentially be extended to color encodings. However, applying these encodings to color is complex because of the nonlinear relationship between color and the human eye's spectral sensitivity [4]. We illustrate the potential of our transform for heatmaps in Fig 8, but a more detailed investigation of these properties applied to color encodings is beyond the scope of this study. Additionally, each of these visualization properties could be formally defined by mathematical relationships between fold changes and transformed outputs, but a more rigorous exploration of these properties is reserved for future research.

We evaluate these properties for linear plots, log plots, and MAD plots (summary found in Fig 1A). We refer to raw log plots as those with the raw log-transformed axis tick marks and labeled log plots as those with the tick marks back-transformed to the original fold change values (e.g. inner y-axis labels of Fig 1D compared to outer labels, respectively). The same notation is used for MAD plots. MAD plots denote any type of plot that uses the MAD-FC transformed fold changes for one of its visual encodings. Raw MAD plots display the raw MAD-FC transformed fold changes, while labeled MAD plots have the axis tick marks back-transformed to the original fold change units. In theory, these plot types are not mutually exclusive if both axes are included, but we discourage this approach because multiple axes can make interpretation more difficult.

A linear plot of fold change displays raw fold change values with a linear transform (Fig 1C). A linear plot of fold change is readable because extracting coordinates from a linear axis tick mapping can be done based on the relative position between tick marks, and no conversion is required since the axis tick marks in a linear plot are the original fold change values. Positive fold changes in a linear plot are proportional since they have a linear relationship to fold change units (Fig 1C, upper right quadrant), while negative fold changes are not proportional (lower left quadrant). This type of plot does not have symmetry since negative fold changes are compressed between [0, 1] while positive fold changes span between $(1, \infty)$. Linear plots of fold change have medium dynamic range because they use a linear axis tick label mapping (see illustrative example in Fig 1B, and Fig 8 for an example using real data).

Raw log plots of fold change use axis tick labels in log units (Fig 1D, inner y-axis labeled $Log_2$(FC)). Extracting the value of a datapoint with this plot is simple because it can be linearly interpolated between axis tick marks (log units are linearly mapped to the axis scale). However, an observer must reverse the log transform to convert the datapoint to the original fold change value, which is a nontrivial process, and even more so for a collection of points. Therefore, a raw log plot is only partially readable. In contrast, a labeled log plot (Fig 1D, outer y-axis labeled FC) has a nontrivial process for extraction because the datapoint value cannot be estimated based on linear interpolation between axis tick marks (back-transformed tick marks are not linearly mapped to the axis scale since the underlying scale is not linear). A labeled log plot has simple conversion because the original fold change values are extracted from the plot directly. Log plots of both types do not have proportionality because of the nonlinearity of the log transform. Log plots are symmetrical, and the log scale gives them a high dynamic range.

A raw MAD plot (that displays the raw output of the MAD-FC transform) has the negative fold changes stretched out to match the scale of the positive fold changes (Fig 1E, inner y-axis labeled MAD-FC). Extracting the value of a datapoint with this plot is simple because the fold change value can be directly interpolated from the axis tick marks. However, the conversion process is nontrivial to obtain fold changes from datapoints since the MAD-FC transform must be reversed. Raw MAD plots are therefore partially readable. In contrast, a labeled MAD plot (Fig 1E, outer y-axis labeled FC) is fully readable because linear interpolation is used for datapoint extraction and axis tick labels use raw fold change units, with no conversion required. MAD plots are proportional because negative fold changes are distorted to match the proportionality of positive fold changes. MAD plots are symmetrical by design and have a medium dynamic range.

While raw log plots and labeled log plots have a trade-off in readability between extraction and conversion (Fig 1A), we note that there is no such trade-off between raw MAD plots and labeled MAD plots. We therefore strongly recommend to never use raw MAD plots to visualize fold change data, because the raw MAD-FC transform units will reduce the readability of the plot and potentially confuse the observer.

## Methods

### MAD-FC transform

We start with a fold change dataset with $n$ measurements that are positive real numbers. To produce a visualization fold change with readability, proportionality, and symmetry, we will perform two transforms on the raw data and then reverse the transforms on the axis tick labels. For this explanation, we use a test dataset comprised of pairs of positive and negative fold changes of the same magnitude (Fig 2A). An actual dataset does not need matching pairs of datapoints; we use these to illustrate how symmetry is maintained with the transform. Notice that reversing the direction of a fold change is simply the reciprocal of its value (i.e., $f(x) = 1/x$).

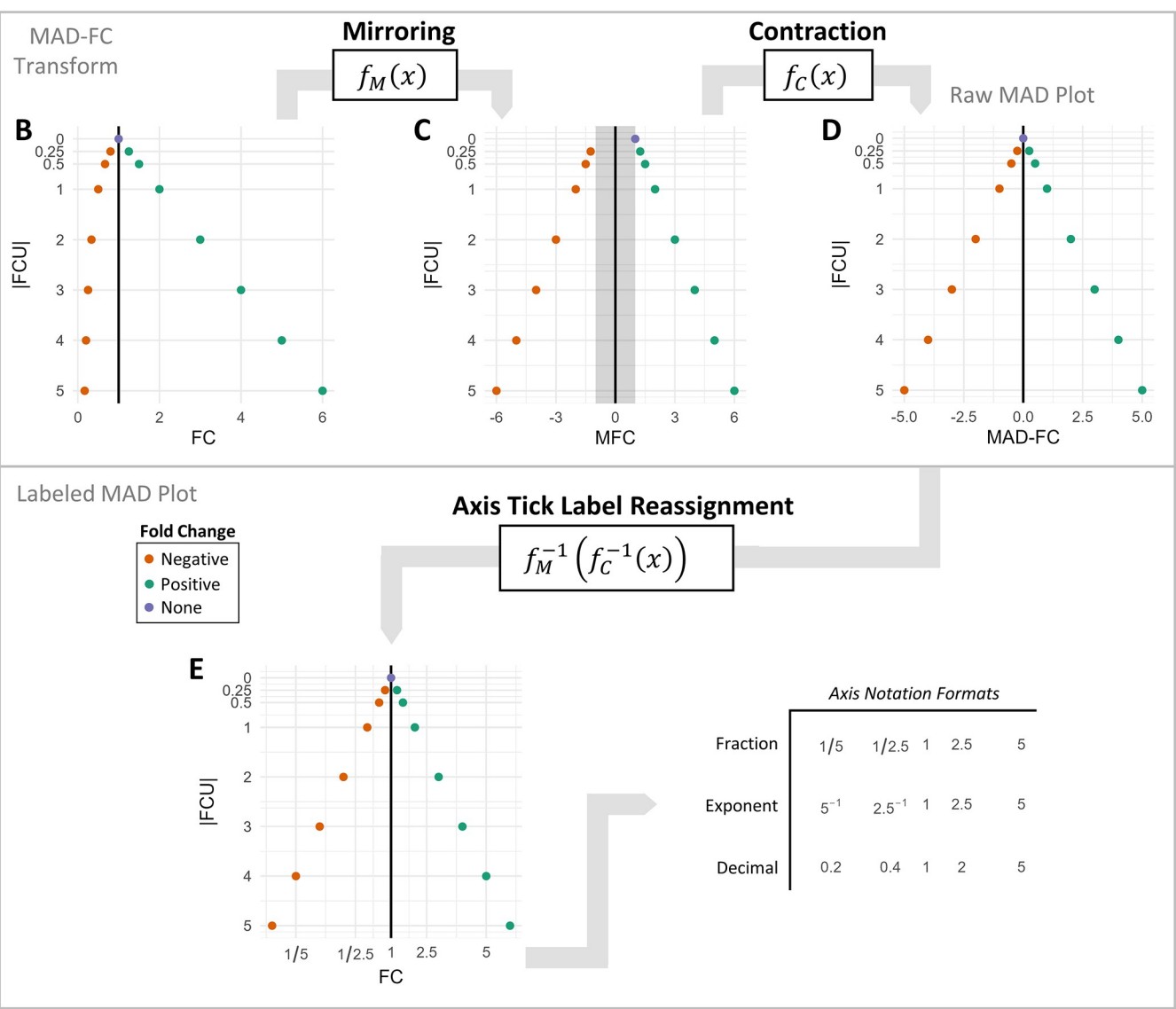

**Fig 2. Illustration of MAD-FC transform and plot.** (**A**) Table of fold change datapoints that pair negative fold changes (–) with their corresponding positive fold changes (+), along with a fold change of 1 denoting the point of no change (NC). (**B**) Plot of fold change datapoints in a linear scale, with negative fold changes compressed between [0, 1] (datapoints from FC column). (**C**) Fold change values with a mirror transform applied ($f_M$) to the negative fold changes to stretch their position to match the corresponding positive fold changes (grey rectangle denotes undefined region between [–1,1], datapoints from MFC column). (**D**) A contraction transform ($f_C$) pulls both positive and negative fold changes 1 unit closer to zero, eliminating the undefined region, but leaving fold change labels shifted 1 unit from their original value (datapoints from MAD-FC column). (**E**) The transforms in (C) and (D) are reversed on the axis tick labels to annotate the datapoints with their actual fold change value. Plot (D) represents a raw MAD plot while the transform reversal in step (E) represents a

labeled MAD plot. A labeled MAD plot can be annotated with axis tick marks formatted as fractions, exponents, or decimals. A labeled MAD plot has datapoints identical in value to the original fold change measurements, but they are spatially distorted to achieve symmetry and proportionality. Column definitions for (**A**): |FCU|, absolute fold change units; Direction, whether fold change datapoint is negative/decreasing (−) or positive/increasing (+), or no change (NC); FC: fold change value; MFC, fold change value after mirror transform; MAD-FC, fold change value after mirror and contraction transform.

In order for a plot of fold changes to be symmetric, these pairs of points must become equidistant to the point of no change.

Based on the table in Fig 2A, we can define a transform that stretches the negative fold change values to match the spacing of the corresponding positive fold changes. We use the same transform that is used to reverse the direction of a fold change and then multiply by negative one. Since we only want to transform negative fold changes, we define a case equation that only transforms negative fold change measurements. We denote this mirror transform as $f_M$, where

$$f_M(x) = \begin{cases} x & x \geq 1 \\ -\dfrac{1}{x} & 0 < x < 1 \\ undefined & otherwise \end{cases}. \tag{2}$$

This transform shifts all negative fold changes to be symmetrically spaced from zero to their corresponding positive fold changes (MFC column in Fig 2A, visualized in Fig 2C). But this transform leaves a discontinuous region between [−1,1] where no fold change values can exist. This would give a misleadingly large spatial distance between negative and positive fold changes close to one (grey region in Fig 2C) and for interval estimates that cross this region. We correct for this by translating all datapoints closer to zero by one unit, defined as a contraction transform $f_C$:

$$f_C(x) = \begin{cases} x - 1 & x \geq 1 \\ x + 1 & x < -1 \\ undefined & otherwise \end{cases}. \tag{3}$$

This transform removes the discontinuous region, but the datapoint values no longer reflect the actual fold change values because they are shifted one unit (MAD-FC column in Fig 2A, visualized in Fig 2D). Eqs (2) and (3) represent the MAD-FC transform, and visualizing this output would represent a raw MAD plot (Fig 3D). If we wish to produce a labeled MAD plot, we will now reverse both transforms on the axis tick labels so that the datapoint's value can be read from the labels. We will perform this label reassignment by reversing the two transforms we performed on the data (i.e., $f_C(x)$ and $f_M(x)$) on the axis tick labels. We first reverse the transform $f_C$ with the function $f_C^{-1}$:

$$f_C^{-1}(x) = \begin{cases} x + 1 & x \geq 0 \\ x - 1 & x < 0 \end{cases}. \tag{4}$$

This transform changes the point of no change of the plot from 0 back to 1 (since the first case in $f_C^{-1}$ includes zero). We now reverse the mirrored fold change transform $f_M$ with $f_M^{-1}$,

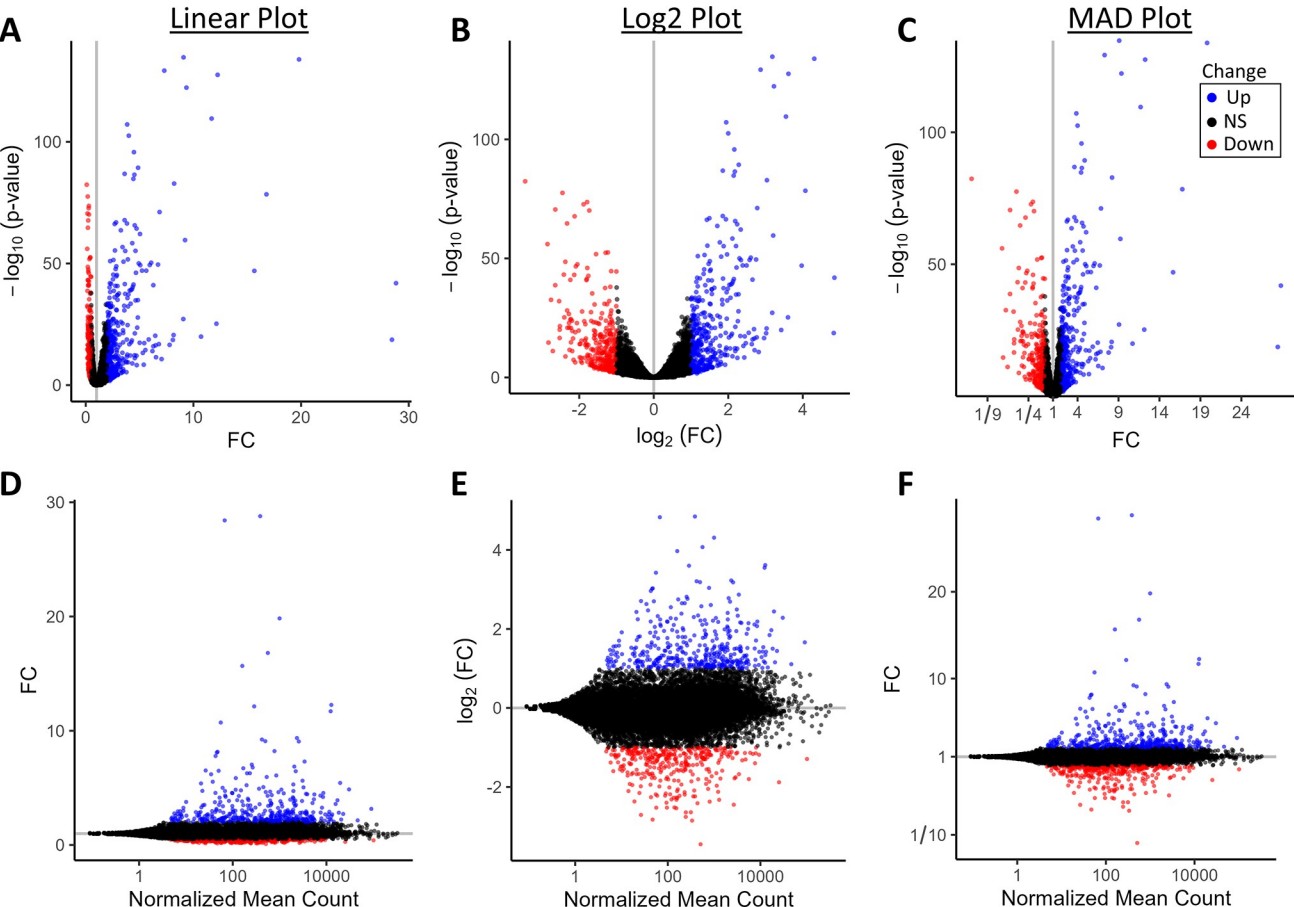

**Fig 3. Comparison of log, linear, and MAD fold change plots for RNA-Seq data.** Volcano plots of p-value versus (**A**) linear, (**B**) log, and (**C**) MAD fold change. MA plots using (**E**) log2, (**F**) linear, and (**G**) MAD fold change versus normalized mean count. Datapoints are annotated as significantly upregulated (Up, red), significant downregulated (Down, blue), or not statistically significant (NS, black) based on a Wald test adjusted p-value < 0.1 and a fold change greater than ± 1.

where

$$
f_M^{-1}(x) = \begin{cases} x & x \geq 1 \\ -\dfrac{1}{x} & x < -1 \\ undefined & otherwise \end{cases}.
\tag{5}
$$

After these transforms, the original fold change values can be read from the plot. We then can display the resulting negative fold change axis tick labels as fractions (Fig 2E), decimals, or with an exponent. We are left with a visualization that not only retains the readability of a linear plot of fold change, but also exhibits proportionality and symmetry around a fold change of 1.

## Data analysis

The dataset used in Fig 3 is from the airway package [5] available through Bioconductor package in R [6]. Differential gene expression analysis is performed with DSeq2 using the Wald test without a beta prior specified, with log fold change shrinkage used to refine the dispersion estimates. The fold change datapoints used for Fig 4D–4F were estimated directly from the left

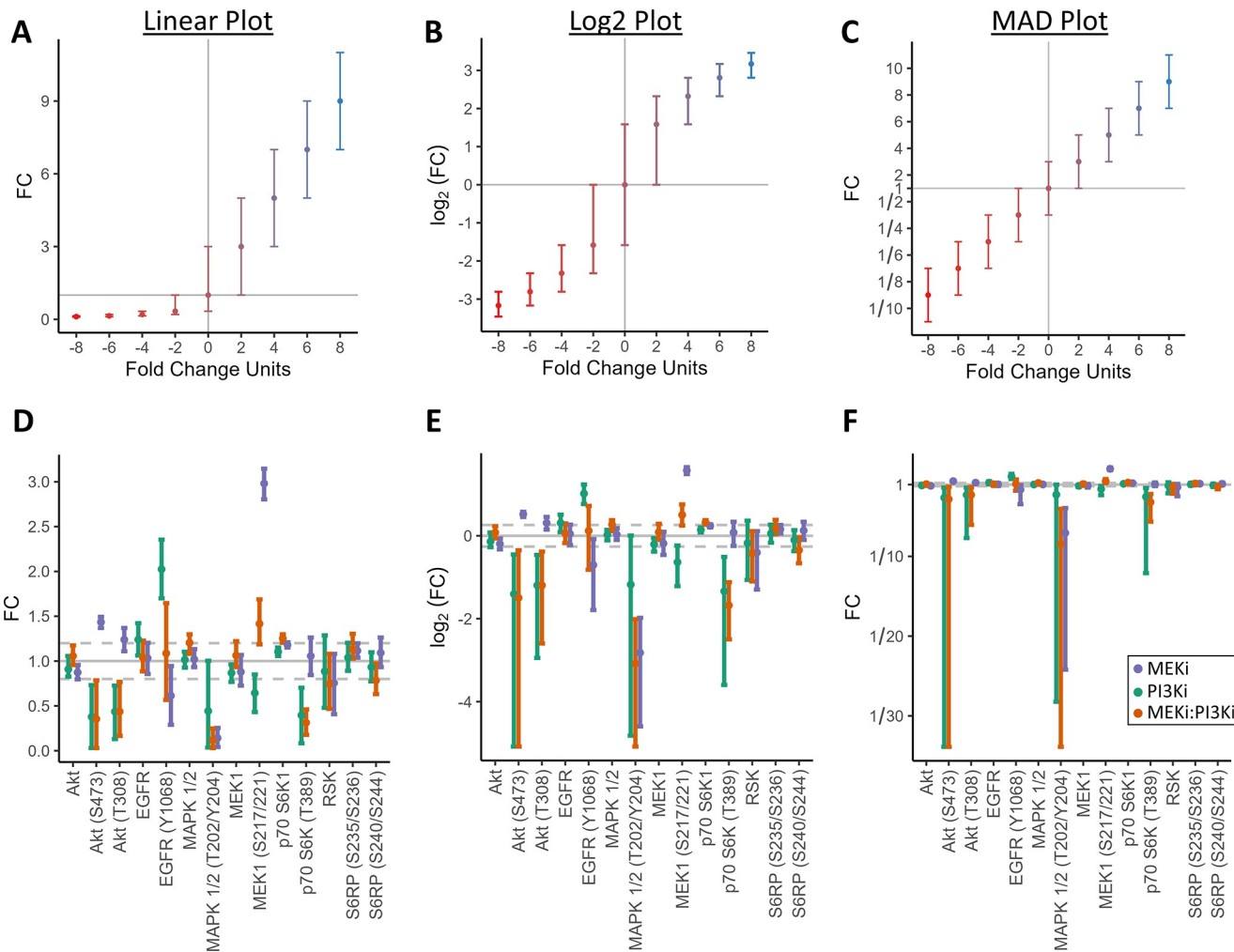

**Fig 4. Comparison between fold change plots with interval estimates.** Fold change interval estimates with the same interval width across groups visualized with a (**A**) linear, (**B**) log2, and (**C**) MAD plot (from a simulated dataset with identical dispersion in fold change units across all groups with interval estimates spanning from -2 to +2-fold change units from the point estimate, error bars are confidence intervals, color gradient used to visually separate groups). Comparison of (**D**) linear, (**F**) log2, and (**F**) MAD fold change plot of protein expression and phosphorylation in HCC1954-P cells treated with either 300nM refametinib (MEKi) or 15nM copanlisib (PI3Ki) alone or in combination (MEKi– 300nM: PI3Ki– 15nM) (error bars are standard deviation).

panel in Fig 6 from the reference publication [7] using WebPlotDigitizer (https://automeris.io/WebPlotDigitizer/). Figs 5 and 6 used gene expression data from the cancer genome atlas using the RTCGA and RTCGA.mRNA packages [8] for five genes of interest. Specifically, RNA expression was extracted from the Breast invasive carcinoma (BRCA), Ovarian serous cystadenocarcinoma (OV) and Lung squamous cell carcinoma (LUSC) datasets. Gene expression measurements were used without any further processing. Fig 7 contains protein expression from a study with quantification for ubiquitin interactors [9]. This data was loaded from the UbiLength dataset contained within the DEP R package [10] and processed with the standard LFQ workflow with default settings, as done in the Introduction to DEP vignette page (please see source code for more information). Fig 8 uses a dataset of RNA expression from leaf tissue of the papaya plant under drought stress [11]. We used preprocessed data made available by the first author (found at https://github.com/sdgamboa/misc_datasets) of the

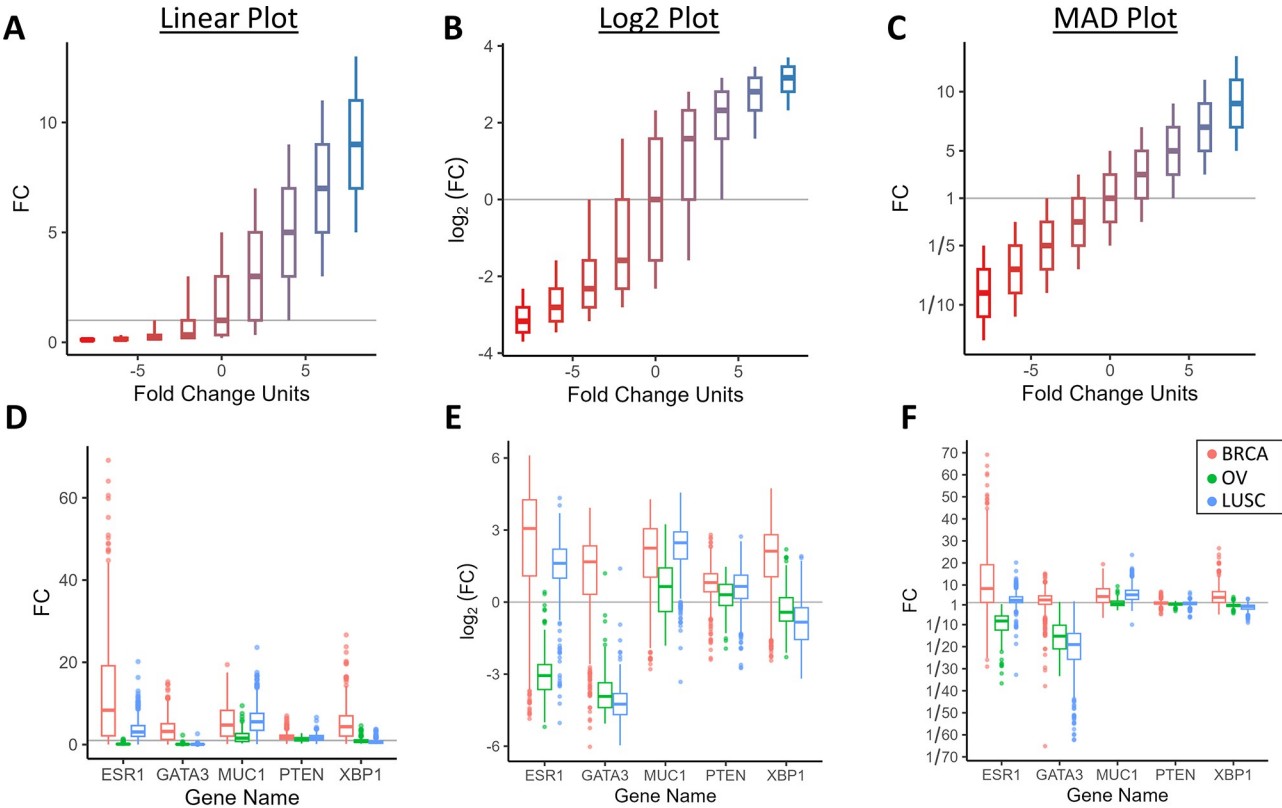

**Fig 5. Comparison between fold change box plots.** Fold change boxplots of simulated data visualized with a (**A**) linear, (**B**) log2, and (**C**) MAD plot (from a simulated dataset with identical dispersion in fold change units across all groups, 2-fold change unit differences between each quartile boundary, color gradient used to visually separate groups). Comparison of (**D**) linear, © log2, and (**F**) MAD plots of mRNA expression of various genes measured from patients with breast invasive carcinoma (BRCA), ovarian serous cystadenocarcinoma (OV) and lung squamous cell carcinoma (LUSC) (datasets from the Cancer Genome Atlas).

source publication [11]. Fold change measurements from the preprocessed dataset were used without any further modifications.

R packages and their version numbers are listed in Table 1.

## Results

Different data visualizations can leave very different impressions of the same dataset and influence the conclusions drawn from the data. We demonstrate the importance of plot types for fold change measurements by comparing log plots, linear plots, and MAD plots of fold change with various biomedical datasets. We highlight the differences in their portrayal of the data, along with the unique advantages of using MAD plots with various plot types, including scatterplots, volcano plots, box plots, violin plots, and heatmaps.

We start with an RNA-Seq dataset that probes differential gene expression from airway smooth muscle cells to investigate the therapeutic mechanisms of glucocorticoid treatment for asthma [5]. Four human airway smooth muscle cell lines were treated with dexamethasone and a control, and gene expression was compared between the two study groups using the DESeq2 package. When performing differential gene expression analysis, it is common to visualize the data with a volcano plot and MA plot. Volcano plots are scatterplots that visualize statistical significance versus effect size and enable the quick identification of genes that exhibit both high statistical significance and large fold change [16].

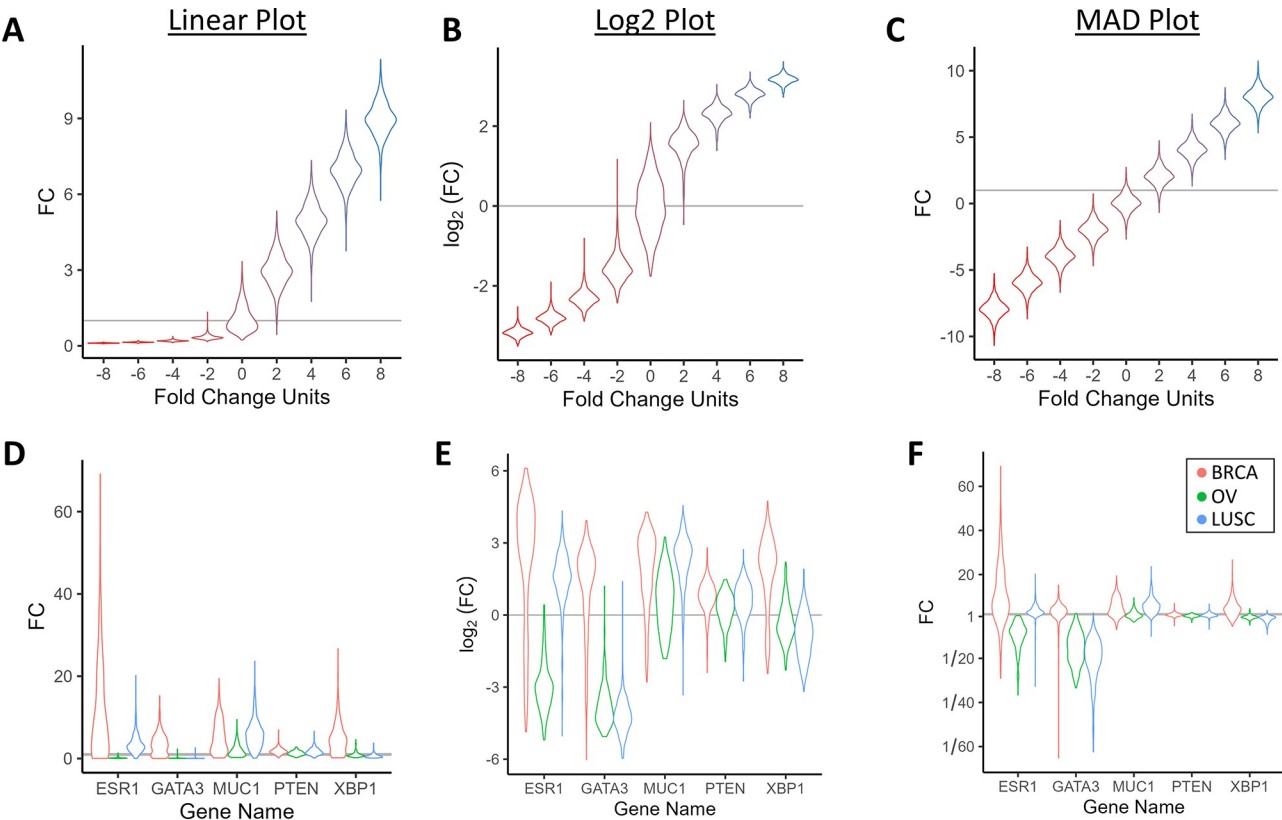

**Fig 6. Comparison between fold change violin plots.** Fold change violin plots with the same dispersion of measurements across groups visualized with a (**A**) linear, (**B**) log2, and (**C**) MAD plot (from a simulated dataset with identical dispersion in fold change units across all groups, color gradient used to visually separate groups). Comparison of violin plots with (**D**) log, (**F**) linear, and (**F**) MAD fold change of mRNA expression of various genes measured from patients with breast invasive carcinoma (BRCA), ovarian serous cystadenocarcinoma (OV), and lung squamous cell carcinoma (LUSC) (data from the Cancer Genome Atlas).

Fig 3A–3C show volcano plots with linear, log, and MAD transforms of fold change for this dataset. The linear plot facilitates comparing the magnitude of positive fold change datapoints, but the negative fold changes are compressed between 0 and 1 (Fig 3A). While the log plot (Fig 3B) has symmetry and facilitates comparisons between positive and negative fold changes, the values of fold change datapoints are not proportional to their distance to the point of no change. The MAD plot of fold change allows for comparison between positive and negative fold changes symmetrically, along with comparing the magnitudes proportionally between fold changes of the same direction (Fig 3C). This type of plot is often used to prioritize candidate genes for further investigation. While the value of each individual datapoint can be read from a log plot, the spatial distribution of the point cloud gives a distorted summary when compared to their actual fold change values. The log plot gives a potentially misleading impression that there are many genes that are reasonably close to the max datapoint value for fold change, but the MAD plot highlights most of these candidates are less than 50% of the max fold change observed in the dataset. This is an important consideration when deciding which gene candidates should be prioritized for follow up studies.

MA plots are used to compare fold change of differential gene expression versus the average count of the same gene between both groups (essentially a Bland-Altman analysis for fold change data with two study groups [17]). MA plots are used to highlight systematic bias or highly differentially expressed genes. We produce MA plots with linear, log, and MAD

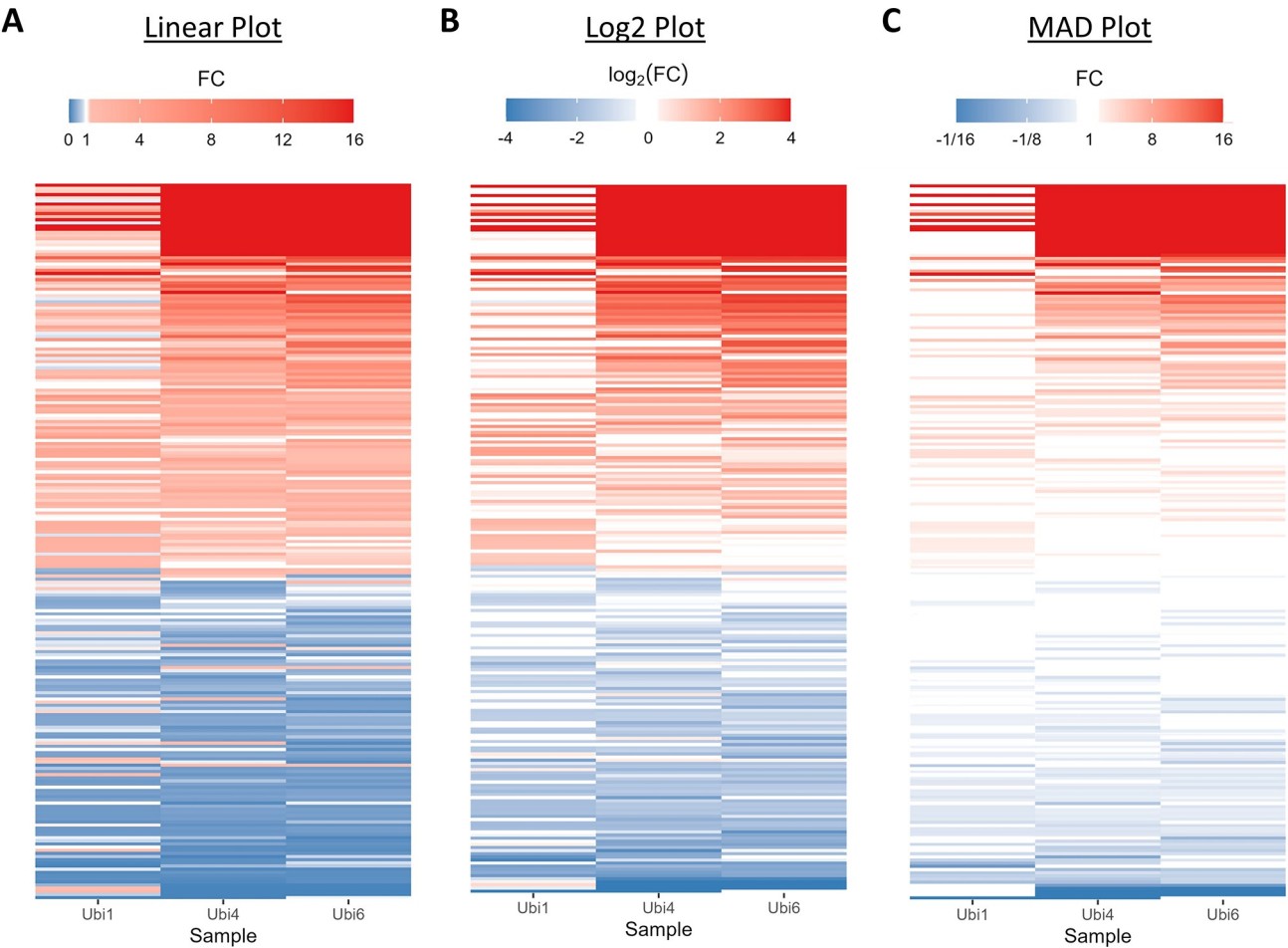

**Fig 7. Comparison of heatmaps with different encodings between fold change and color.** Comparison of heatmaps with a (**A**) linear, (**B**) log2, and (**C**) MAD-FC transformed color mapping of differential expression of proteins that interact with ubiquitin, a regulatory protein found in all eukaryotic organisms. The rows are proteins that are identified as Ubiquitin-protein interactors while the columns are experiment groups that represent ubiquitins of specific chain lengths (linear mono (Ubi1), tetra (Ubi4), and hexa-ubiquitin (Ubi6)). Experiments were performed on HeLa cells in vitro and expression for each group are averaged across three replicates. Note: Color scale mapped to fold changes after each of the transforms.

transforms (Fig 3D–3F) of the same dataset used in Fig 3A–3C. The advantages of the MAD MA plot are the same as with volcano plots- the values of the datapoints are presented without the spatial distortion found with the log and linear plots.

The MAD-FC transform is especially useful for visualizing the sample distribution and uncertainty associated with fold change measurements (whether the standard deviation, standard error, quartile, distribution, confidence interval, credible interval, or support interval). To illustrate this, we produce a simulated dataset of fold change measurements with a 95% confidence interval that extends 2-fold change units above and below the point estimate for each study group. These study groups all have identical interval widths in fold change units, yet the MAD plot is the only visualization where this consistency is apparent (Fig 4A–4C). Both the log and linear fold change plots heavily distort the interval width, making it impossible to perceive that all the study groups have the same confidence interval widths. The MAD plot not only preserves the proportional distance of each datapoint to the point of no change, but also preserves the interval estimate width regardless of the value of the fold change measurement. This key behavior is apparent with a dataset measuring protein expression and

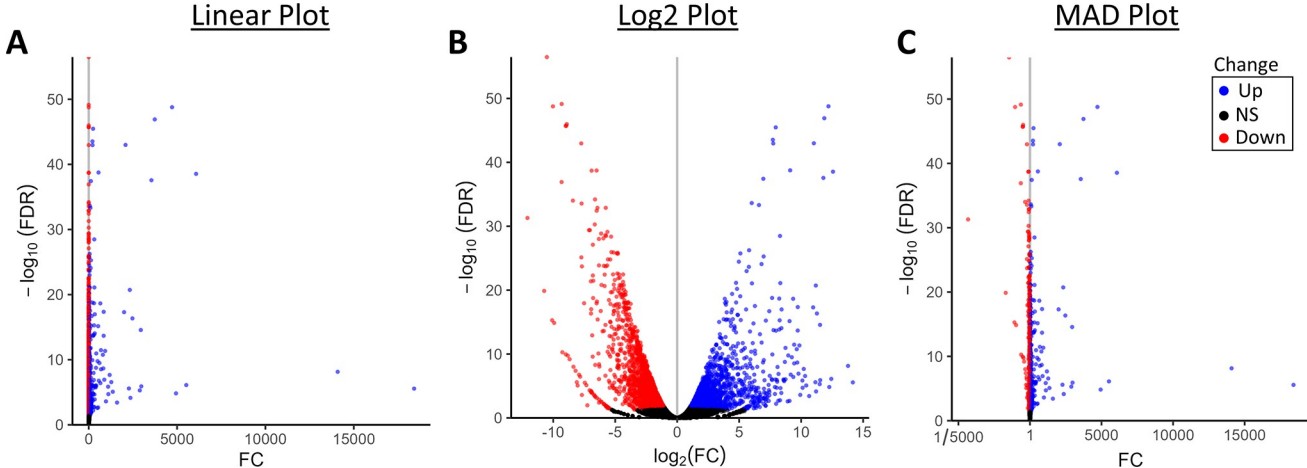

**Fig 8. Comparison of volcano plots with a dataset of fold change values with high dynamic range.** Comparison of heatmaps with (**A**) linear, (**B**) log2, and (**C**) MAD fold change color mapping of differential gene expression of papaya leaves tissue after drought stress compared to control. Datapoints are annotated as significantly upregulated (Up, blue), significant downregulated (Down, red), or not statistically significant (NS, black) based on an FDR < 0.05 and a fold change greater than ± 1.

phosphorylation in HER2-positive breast cancer cell lines treated with the MEK inhibitor refametinib [7] (Fig 4D–4F). The linear visualization of fold change (Fig 4D) used in the cited publication makes it easy to mistakenly conclude that the standard deviation appears approximately consistent across measurements and the negative fold change measurements are smaller in magnitude than the largest positive fold change measurement. However, when the same data is viewed by log plot (Fig 4E) or MAD plot (Fig 4F), the negative fold changes are revealed to have much larger standard deviations than the other measurements and they are larger in magnitude than the largest positive fold change datapoint. The advantage with the MAD plot over the log plot is that the width of the bounds for standard deviation can be compared directly and proportionally between groups regardless of their distance from the point of no change.

The distortions exhibited by log and linear plots are even more pronounced in boxplots. A simulated dataset visualized with boxplots with the sample median swept from 1/9 to 9-fold changes, with quartiles evenly spaced 2-fold changes between themselves, shows that not only the visualized width of the boxplots changes dramatically, but also the relative widths of quartiles within a single boxplot can become heavily distorted depending on the plot used

**Table 1. Packages used in code repository.**

| Package | Version | Reference |
|---|---|---|
| DESeq2 | 1.40.2 | [2] |
| EnhancedVolcano | 1.18.0 | [12] |
| tidyverse | 2.0.0 | [13] |
| BiocManager | 1.30.22 | [14] |
| magrittr | 2.0.3 | [15] |
| Bioconductor | 1.30.22 | [14] |
| airway | 1.20.0 | [5] |
| RTCGA | 1.30.0 | [8] |
| RTCGA.mRNA | 1.28.0 | [8] |
| DEP | 1.23.0 | [10] |

(Fig 5A–5C). This characteristic gives a false impression that the 4th and 6th study group have skewed distributions in fold change within the log plot (Fig 5A) when they are in fact symmetrical (Fig 5C). With boxplots of mRNA expression of several genes measured from tumor tissue of several cancer types, the log, linear, and MAD plots appear as if they represent entirely different datasets (Fig 5D–5F). Log plots of fold change exaggerate graphical features that are close to zero and compress those that are further away. Linear plots of fold change heavily distort any features that extend into the region of negative fold change.

The same trends are observed when comparing violin plots between each of the three visualizations. With a simulated dataset of fold change distributions uniformly translated to different fold change values while maintaining the same degree of dispersion in fold change units (Fig 6A–6C), only the MAD plot reveals that each of these study groups have the same distribution shape, while the log and linear plots heavily distort the distribution depending on the fold change values. When displaying the same dataset as Fig 5D–5F with violin plots, the overall appearance and trends of the datasets are again dramatically different for each of the visualizations (Fig 6D–6F).

The MAD-FC transform can also be useful for mapping fold change data to color gradients. A comparison of fold change heatmaps for differential protein expression (Fig 7A–7C) reveals that MAD-FC emphasizes measurements with the largest fold change values in the dataset. Mapping logarithmically transformed fold changes to a color gradient makes it more difficult to clearly discern the largest fold change measurements (dataset from a study measuring protein expression of Ubiquitin-protein interactors [9]).

As mentioned previously, using the MAD-FC transform to visualize fold changes may not be useful with datasets with a dynamic range of more than 8 units in a log2 scale. Although most bioinformatics datasets do not satisfy this condition, there are exceptions. One such example is a dataset from a study investigating gene expression in response to drought stress in the papaya plant [11] (Fig 8A–8C). This dataset spans ±15 units on the log2 scale. While the log plot spreads out the points for effective visual inspection, the linear and MAD plots compress nearly the entire dataset onto the point of no change, limiting the usefulness of such visualizations. For this specific data set, a log-transform would be more useful for most applications.

## Discussion

Here we propose a transform that enhances the usefulness of a linear visualization of raw fold changes. A major shortcoming of linear plots of fold change is that negative fold change values are compressed from [0, 1] and cannot be compared to positive fold change values by their spatial position. This limitation hinders acquiring a holistic summary of fold change values.

Log plots allow the comparison of fold changes from both directions because positive and negative fold changes are positioned symmetrically about the origin. Yet the proportional relationship between fold change value and spatial position is lost for log plots. While individual points can be read from a log plot with those familiar with the log scale spacing of values between axis tick labels, it is difficult to identify linear trends from point clouds in a log scale. In contrast, MAD plots are designed to maintain readability, symmetry, and proportionality. This combination of characteristics allows spatial position to be used as a proportional encoding for fold change value regardless of direction or magnitude of the measurement. As a consequence, MAD plots are especially useful for visualizing the distribution, quartiles, and interval estimates of fold change measurements since these visual features are not distorted in a spatially dependent manner as in log and linear plots of fold change.

We enhance linear visualization of fold change with a transform strategy that is borrowed from labeled log plots, where the axis tick labels undergo a reverse transform to display the

original fold change values. While this visualization lacks the high dynamic range found in log plots, in many applications comparing fold change across 2.5 orders of magnitude (8 units on the log 2 axis) is sufficient to identify interesting datapoints. This new visualization may be a useful tool for more intuitively summarizing fold change values. Such visualizations could perhaps be used for other applications than what is shown here, such as meta-analysis techniques [18] and comparing effect size across broadly related experiments as we have done in previous studies [19–21].

## Acknowledgments

We thank the reviewers for their insightful feedback that greatly improved the quality of this manuscript.

## Author Contributions

**Conceptualization:** Bruce A. Corliss.

**Data curation:** Bruce A. Corliss.

**Formal analysis:** Bruce A. Corliss, Francis P. Driscoll.

**Investigation:** Bruce A. Corliss.

**Methodology:** Bruce A. Corliss, Francis P. Driscoll.

**Software:** Bruce A. Corliss.

**Supervision:** Philip E. Bourne.

**Validation:** Bruce A. Corliss.

**Visualization:** Bruce A. Corliss.

**Writing – original draft:** Bruce A. Corliss.

**Writing – review & editing:** Bruce A. Corliss, Yaotian Wang, Heman Shakeri, Philip E. Bourne.

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
