## [Editor Report · Decision Letter 0]

5 May 2023

PONE-D-23-12801MAD-FC: a fold change visualization with readability, proportionality, and symmetryPLOS ONE

Dear Dr. Corliss,

Thank you for submitting your manuscript to PLOS ONE. We invite you to submit a revised version of the manuscript that addresses the point raised below.

Kind regards,

Stephen R. Piccolo

Academic Editor

PLOS ONE

Journal Requirements:

Additional Editor Comments (if provided):

I'm sending this back to you (before sending it out for review) to ask that you submit a revised version with higher resolution figures. I'm not asking for the figures to be at extremely high resolution. However, the resolution is so low in the current submission that the figures look quite grainy, so it is difficult to decipher some of the patterns and text. Because visualization is the whole point of this paper, it will be important to make sure the reviewers will be able to see the figures well.
---

## [Author Response · Author response to Decision Letter 0]

11 May 2023

Figures were reproduced at higher resolution (600 DPI) and processed through Plos One's PACE tool. Unfortunately, the figures embedded within the PDF are still compressed and at low resolution, but there is a download link at the top of each page where the original high-resolution files can be downloaded. I talked to the editorial office, and they said there is no way I can boost the resolution of the embedded images within the PDF.

---

## [Decision Letter · Decision Letter 1]

31 Jul 2023

PONE-D-23-12801R1MAD-FC: a fold change visualization with readability, proportionality, and symmetryPLOS ONE

Dear Dr. Corliss,

Thank you for submitting your manuscript to PLOS ONE. After careful consideration, we feel that it has merit but does not fully meet PLOS ONE’s publication criteria as it currently stands. Therefore, we invite you to submit a revised version of the manuscript that addresses the points raised during the review process.

We look forward to receiving your revised manuscript.

Kind regards,

Stephen R. Piccolo

Academic Editor

PLOS ONE

Journal Requirements:

Additional Editor Comments:

Two reviewers provided feedback on your manuscript. Both had positive things to say. One of them provided a detailed list of suggestions for improving the manuscript. I am inclined to trust that these recommendations will indeed improve the manuscript. But if you can provide convincing justification that some are unnecessary, feel free to make that case.

Reviewers' comments:

Reviewer's Responses to Questions

**Comments to the Author**

1. If the authors have adequately addressed your comments raised in a previous round of review and you feel that this manuscript is now acceptable for publication, you may indicate that here to bypass the “Comments to the Author” section, enter your conflict of interest statement in the “Confidential to Editor” section, and submit your "Accept" recommendation.

Reviewer #1: (No Response)

Reviewer #2: (No Response)

2. Is the manuscript technically sound, and do the data support the conclusions?

Reviewer #1: Yes

Reviewer #2: Partly

3. Has the statistical analysis been performed appropriately and rigorously? 

Reviewer #1: Yes

Reviewer #2: No

4. Have the authors made all data underlying the findings in their manuscript fully available?

Reviewer #1: Yes

Reviewer #2: No

5. Is the manuscript presented in an intelligible fashion and written in standard English?

Reviewer #1: Yes

Reviewer #2: Yes

6. Review Comments to the Author

Reviewer #1: The authors present a new data transformation method that aims to improve data visualization and interpretation. Large datasets with fold-change data are often presented on a log scale for its large dynamic range. This scale, however, has the disadvantage that it is not proportional. The new approach, MAD-FC, yield a proportional scale which improves readability and, as a consequence, interpretation.

In my opinion, the work is original and relevant. I have no comments on the content of the article and I only want to encourage the authors to make the process as easy and transparent for potential future users. It would be nice to see people try and adapt this method.

The code to produce all figures is available and that’s great start. Also, the procedure is detailed in the manuscript, step-by-step, but instructions on how to do this on ones own data can be improved.

So, for a true novice that wants to implement this, an excel sheet with formula’s (if possible), a video tutorial or a more detailed walk-through in R (or even better R markdown) may be helpful.

These are suggestions only and should not be regarded as requirements that need to be met before publication.

Reviewed by Joachim Goedhart (University of Amsterdam, the Netherlands).

Reviewer #2: Please see attachment. (The website wants me to include at least 100 characters of text here and isn't letting me submit the form until I finish this sentence...? Okay, now it's good.)

7. PLOS authors have the option to publish the peer review history of their article (what does this mean?). If published, this will include your full peer review and any attached files.

Reviewer #1: **Yes: **Joachim Goedhart (University of Amsterdam, the Netherlands)

Reviewer #2: **Yes: **Marcus W. Fedarko

---

## [Author Response · Author response to Decision Letter 1]

18 Oct 2023

>> Note: Author responses to each query being with “>>”.

PONE-D-23-12801R1

MAD-FC: a fold change visualization with readability, proportionality, and symmetry

PLOS ONE

Dear Dr. Corliss,

Thank you for submitting your manuscript to PLOS ONE. After careful consideration, we feel that it has merit but does not fully meet PLOS ONE’s publication criteria as it currently stands. Therefore, we invite you to submit a revised version of the manuscript that addresses the points raised during the review process.

We look forward to receiving your revised manuscript.

Kind regards,

Stephen R. Piccolo

Academic Editor

PLOS ONE

Journal Requirements:

Additional Editor Comments:

Two reviewers provided feedback on your manuscript. Both had positive things to say. One of them provided a detailed list of suggestions for improving the manuscript. I am inclined to trust that these recommendations will indeed improve the manuscript. But if you can provide convincing justification that some are unnecessary, feel free to make that case.

Reviewers' comments:

>> We want to thank both reviewers for their constructive feedback that enhanced the quality of the manuscript.

Reviewer's Responses to Questions

Comments to the Author

1. If the authors have adequately addressed your comments raised in a previous round of review and you feel that this manuscript is now acceptable for publication, you may indicate that here to bypass the “Comments to the Author” section, enter your conflict of interest statement in the “Confidential to Editor” section, and submit your "Accept" recommendation.

Reviewer #1: (No Response)

Reviewer #2: (No Response)

2. Is the manuscript technically sound, and do the data support the conclusions?

Reviewer #1: Yes

Reviewer #2: Partly

3. Has the statistical analysis been performed appropriately and rigorously?

Reviewer #1: Yes

Reviewer #2: No

4. Have the authors made all data underlying the findings in their manuscript fully available?

Reviewer #1: Yes

Reviewer #2: No

5. Is the manuscript presented in an intelligible fashion and written in standard English?

Reviewer #1: Yes

Reviewer #2: Yes

6. Review Comments to the Author

Reviewer #1: The authors present a new data transformation method that aims to improve data visualization and interpretation. Large datasets with fold-change data are often presented on a log scale for its large dynamic range. This scale, however, has the disadvantage that it is not proportional. The new approach, MAD-FC, yield a proportional scale which improves readability and, as a consequence, interpretation.

In my opinion, the work is original and relevant. I have no comments on the content of the article and I only want to encourage the authors to make the process as easy and transparent for potential future users. It would be nice to see people try and adapt this method.

The code to produce all figures is available and that’s great start. Also, the procedure is detailed in the manuscript, step-by-step, but instructions on how to do this on ones own data can be improved.

So, for a true novice that wants to implement this, an excel sheet with formula’s (if possible), a video tutorial or a more detailed walk-through in R (or even better R markdown) may be helpful.

These are suggestions only and should not be regarded as requirements that need to be met before publication.

>> You make a great point that this work can only be useful if it is accessible. Although optional, we went ahead and started the process by making a website with vignettes in RMarkdown for each of the examples in the manuscript (link added to methods section). In the future, we hope to make an R package, and then expand to other programming languages.

Reviewed by Joachim Goedhart (University of Amsterdam, the Netherlands).

Reviewer #2: Please see attachment. (The website wants me to include at least 100 characters of text here and isn't letting me submit the form until I finish this sentence...? Okay, now it's good.)

7. PLOS authors have the option to publish the peer review history of their article (what does this mean?). If published, this will include your full peer review and any attached files.

Do you want your identity to be public for this peer review? For information about this choice, including consent withdrawal, please see our Privacy Policy.

Reviewer #1: Yes: Joachim Goedhart (University of Amsterdam, the Netherlands)

Reviewer #2: Yes: Marcus W. Fedarko

Review of Corliss et al., 2023 (“MAD-FC: a fold change visualization with readability, proportionality, and symmetry”)

Corliss et al. describe MAD-FC, a transform for fold change values designed to produce easier-tointerpret visualizations (compared to visualizations of raw fold changes and visualizations of log fold changes). The authors outline four desirable properties of fold change visualizations (readability, proportionality, symmetry, and high dynamic range), argue that plots using MAD-FC-transformed fold changes exhibit readability, proportionality, and symmetry, and present various examples of these plots compared with plots of raw and log fold changes.

Visualizations of raw fold changes do not represent negative fold changes well: although positive fold changes fill the interval of (1, ∞), negative fold changes are constrained to the interval [0, 1). The scale of fold changes in a MAD-FC plot is adjusted to account for this, allowing for negative and positive fold changes of equal magnitude to be symmetrically placed on both sides of 1 (the point marking “no change”). Log plots of fold changes also exhibit symmetry, since log(A/B) = -log(B/A) for all positive A and B; however, unlike log plots, MAD-FC plots preserve fold changes’ linearity.

I believe that the proposed methods have merit and could be widely useful. However, the paper has some serious issues that should be addressed before publication. Among other major and minor criteria, some important issues I see include:

• I am not convinced that the four described properties of fold change visualizations are well defined;

• The methods detailing the transform are not clearly written;

• The paper lacks details describing how Figures 3–7 were created, and does not provide information about the locations of the corresponding data;

• The paper uses inconsistent terminology and has a nontrivial amount of typographic errors; and

• I believe that the paper places too much of an emphasis on axis ticks and labels rather than on the MAD-FC transform itself.

>> We think that all of these are great points; we discuss specific corrections for each of them below. We rewrote much of the introduction and rethought our definitions used for the visualization properties. We thank the reviewer for providing extensive feedback that we believe significantly increased the quality of the manuscript.

Below are my comments. Please note that I will refer to page numbers in the draft PDF (spanning 43 pages): specifically, I will refer to pages in the first version of the manuscript within this PDF (within the range of pages 7–28).

Major points

1. The words “graph,” “plot,” and “chart” are all used (apparently interchangeably) throughout the manuscript (for example, the abstract says “plots of fold change” and “fold change charts” in multiple places).

The third reference that the authors cited (Midway 2020) agrees that these terms are often used interchangeably (see Box 1 of that paper). For the sake of clarity, I recommend sticking to only one of these terms throughout the manuscript, since the definition of “chart” given in the Background section (page 9) makes it seem as if charts are distinct, somehow, from plots and graphs. In particular, I suggest replacing “graph” and “chart” with “plot” (since by my count “graph” is used 4 times in the paper, “chart” is used 5 times, and “plot” is used around a hundred times), although I leave the decision up to the authors.

>>> Great point, we changed all mentions of “graph” and “chart” to plot.

2. Many parts of the paper refer to “log plots” and “linear plots” of fold change, and a recurring implicit claim in the paper (including at least the title, abstract, and introduction) is that MADFC represents a new visualization method, in addition to a new mathematical transform.

I contend that referring to MAD-FC in this way overcomplicates the paper, and that it would be clearer to just refer to MAD-FC as a transform (albeit a transform which has been defined primarily for the purposes of visualization). This is analogous to how we can define a logarithmic transform, but the logarithm (by itself) is not a visualization method: it is a way we can adjust the scale(s) being used in a visualization. We might sometimes refer to “log plots” in shorthand, but these are more precisely described as “plots of log-transformed data.”

Similarly, the authors use the MAD-FC transform in multiple types of “MAD-FC plots”—in ordinary scatter plots (Figure 1F, 2E–G), volcano plots (Figure 3C), box plots (Figure 5C), heatmaps (Figure 7C), etc. It’s subtle (and I understand if the authors have reservations about making this change throughout the entire paper), but I strongly recommend adjusting the paper to move away from the term “MAD-FC plot” in favor of describing the “MAD-FC transform,” the results of which can then be visualized in many different types of plots. (Another fix would be explicitly defining “MAD-FC plot” to mean “any plot that uses MAD-FC-transformed fold changes,” etc.).

When I first began reading through this paper, I assumed that the authors were defining a new type of plot—but the main contribution here (and it is a worthwhile contribution!) is the transform that the authors present, rather than the way this transform is plotted. This is what is worth emphasizing.

>> We followed your recommendation and restructured the paper to place more emphasis on the MAD-FC transform and defined MAD plot to mean any plot that uses MAD-FC transformed fold changes for one of its linear encodings.

3. I propose replacing the term “reference point” with “point of no change,” or something similar. This will make it consistent with the term “fold change from no change” (also, the terms “reference point of no change” and “point of no change” pop up multiple times in the paper anyway).

>> Solid point, changed all wording of “reference point” to “point of no change”.

4. The paper introduces the term “fold change from no change”, but also makes use of the term 

“fold change units” throughout. It seems to me like these are different ways of describing the 

same concept, so I recommend sticking to one of them. (If I’m reading this wrong and they are two separate things, that should be made clearer—I don’t see much reference information online when I search for “fold change units”, so I expect other readers will be similarly confused.)

>> Great point, and your assumption is correct. We removed all mentions of “fold change from no change” and only used “fold change units” (it seemed less awkward to use that form in sentences and axis labels).

5. I think the definitions of “readability” (Background, page 10: “a visualization with readability has a clear and direct mapping between the value of the datapoints and their spatial location”) and “proportionality” (Background, page 10: “a visualization exhibits proportionality if the fold change datapoints going the same direction are proportionally distant from the value denoting no change within the plot”) need some work. I suspect that it might be best to merge “readability” and “proportionality” together, since these terms’ definitions seem to overlap.

1. Arguably, a visualization using a logarithmic scale also has such a clear and direct mapping —the only difference is that it is a logarithmic mapping, and thus can be slightly harder to interpret. You may want to consider renaming “readability” to “linearity,” or saying “a visualization with readability has a 

---

## [Decision Letter · Decision Letter 2]

14 Nov 2023

PONE-D-23-12801R2MAD-FC: a fold change visualization with readability, proportionality, and symmetryPLOS ONE

Dear Dr. Corliss,

Thank you for submitting your manuscript to PLOS ONE. After careful consideration, we feel that it has merit but does not fully meet PLOS ONE’s publication criteria as it currently stands. Therefore, we invite you to submit a revised version of the manuscript that addresses the points raised during the review process.

We look forward to receiving your revised manuscript.

Kind regards,

Stephen R. Piccolo

Academic Editor

PLOS ONE

Journal Requirements:

Additional Editor Comments:

Thank you for addressing the reviewers' concerns. The reviewers have looked at the revised version. One reviewer provided a detail list of additional suggestions. I do think that many of these will improve the paper, so I ask you go through them and consider whether you agree. If so, please make those changes. If you do not agree any of the changes, please provide a short response on those items, and I will evaluate them (likely without sending it out again for review).

Reviewers' comments:

Reviewer's Responses to Questions

**Comments to the Author**

1. If the authors have adequately addressed your comments raised in a previous round of review and you feel that this manuscript is now acceptable for publication, you may indicate that here to bypass the “Comments to the Author” section, enter your conflict of interest statement in the “Confidential to Editor” section, and submit your "Accept" recommendation.

Reviewer #1: All comments have been addressed

Reviewer #2: (No Response)

2. Is the manuscript technically sound, and do the data support the conclusions?

Reviewer #1: Yes

Reviewer #2: Yes

3. Has the statistical analysis been performed appropriately and rigorously? 

Reviewer #1: Yes

Reviewer #2: Yes

4. Have the authors made all data underlying the findings in their manuscript fully available?

Reviewer #1: Yes

Reviewer #2: Yes

5. Is the manuscript presented in an intelligible fashion and written in standard English?

Reviewer #1: Yes

Reviewer #2: Yes

6. Review Comments to the Author

Reviewer #1: (No Response)

Reviewer #2: Please see the attached PDF of comments. I feel that the paper has strongly improved since the last version, although I believe a few issues remain that should be addressed or at least acknowledged before publication.

7. PLOS authors have the option to publish the peer review history of their article (what does this mean?). If published, this will include your full peer review and any attached files.

Reviewer #1: **Yes: **Joachim Goedhart

Reviewer #2: **Yes: **Marcus W. Fedarko

---

## [Author Response · Author response to Decision Letter 2]

13 May 2024

Response to Reviewer Comments

Review 1

We thank reviewer 1 for their feedback with the first revision and are happy that they feel we have addressed their suggestions.

Review 2 of Corliss et al., 2023 (“MAD-FC: a fold change visualization with readability, proportionality, and symmetry”)

The authors have done an excellent job improving the paper. The additional rigor and detail introduced in this revision, including the additions to Figure 1 and the new Figure 8, makes the paper much stronger—thank you!

In particular, the revisions made to the four desirable criteria of a fold change visualization (readability, proportionality, symmetry, and high dynamic range) make the foundation of the paper much clearer. I still have some lingering questions about the readability criterion, however, and I have a few ideas that I think would improve the presentation of the other three criteria.

The majority of the remaining suggestions I have now are minor points; many of these are debatable issues with phrasing or inessential suggestions, but there are some remaining clear errors that should be addressed.

In the comments below, I will refer to line numbers in the “clean” copy of the revised manuscript.

>> We thank this reviewer for all of the fantastic feedback!

Major points

1. Definitions used in “Readability”: The new definitions used in the context of readability help illustrate your intent much better—however, I still have a few remaining issues with these terms.

1. The definition of readability given in the introduction (lines 60–69) seems to only describe this in terms of plots where the fold change is used as a spatial encoding, i.e. not including Figure 7 (which shows heatmaps of fold changes).

1. You could probably define some sort of relationship between spatial and color-based encodings that allows us to apply “readability” to the latter, but the introduction section does not seem to do this as of writing.

2. It’s fine if “readability” only applies to spatial plots (I think going very in-depth about how MAD-FC can be used for heatmaps is probably out of scope of this paper). However, I would at least recommend adding a small disclaimer (likely to the introduction) that the descriptions of these four properties are mainly given in the context of spatial plots.

3. This also impacts the other three properties (proportionality, symmetry, and dynamic range) somewhat—however, these three properties seem a bit easier to describe in the context of a color-based encoding like that used in Figure 7. The possible more formal definitions of proportionality and symmetry I mentioned below (see “Introduction, lines 75–83 and 86–88”) may help with such a description? That being said, as stated above and below I do not think these particular issues are essential to resolve before publication.

>> Great points, we added this disclaimer on line 107: “It is important to note that while these properties are explained in the context of spatial encodings, they could potentially be extended to color encodings. However, applying these encodings to color is complex because of the nonlinear relationship between color and the human eye’s spectral sensitivity (4). We illustrate the potential of our transform for heatmaps in Figure 8, but a more detailed investigation of these properties applied to color encodings is beyond the scope of this study.”

2. Distinguishing raw and labeled plots has made the explanation of readability much clearer, I think! The main issue I have with the current presentation is that the paper implies that these are two mutually exclusive options—but, as Figures 1D and 1E show, it is possible to have both types of tick labels in the same plot.

1. Of course, in practice most plots (like Figures 3–8) will only show one type of label, so I think your distinction still makes sense. I will leave the exact way of handling it up to the authors, but I recommend at least adding a small disclaimer somewhere that there doesn’t have to be a “dichotomy” between raw and labeled plots. (Unless I am missing something—or, if you feel that combining labels in this way makes the plot hard to read, then it would be worth discussing that in the text.)

2. I think taking some more time in the introduction section that explains these terms (lines 107–109) to go into more detail would help clarify things.

>> Valid points, we added text the line 116: “In theory, these plot types are not mutually exclusive if both axes are included, but we discourage this approach because multiple axes can make interpretation more difficult.” Also added text to line 150: “While raw log plots and labeled log plots have a trade-off in readability between extraction and conversion (Figure 1A), we note that there is no such trade-off between raw MAD plots and labeled MAD plots. We therefore strongly recommend to never use raw MAD plots to visualize fold change data, because the raw MAD-FC transform units will potentially confuse the read and degrade the readability of the plot.”

2. Evaluating proportionality and symmetry: Introduction, lines 75–83 and 86–88: The descriptions of how to visually evaluate proportionality and symmetry have made this section much clearer—thank you! This is not an essential suggestion, so feel free to skip it if you’d like, but I propose more formally defining the ways in which you evaluate these properties.

1. For example, rather than saying that “Symmetry of a transform can be visually assessed by measuring the distance between synthetically generated pairs of fold changes of opposite direction with the same magnitude”, it would be more correct to provide a mathematical definition of symmetry (e.g. given a transform function f(x), this transform exhibits symmetry if f(x) = -f(1/x)).

1. This accounts for the unlikely case where a transform is only partially symmetric for some reason (e.g. only up to within a certain distance from the point of no change)— such a transform should not be called “symmetric,” and yet the visual assessment method currently described in the paper would result in this transform being called “symmetric” unless the synthetic dataset generated for testing extends far enough.

2. Similarly, instead of describing the process of drawing a line between the largest positive fold change and the point of no change and then checking to see if all positive points fall on this line, you could say that a transform function f(x) is proportional if, for all fold changes x > 1, f(x) = mfU(x) – mfU(1) + f(1) given some real m > 0.

1. At least, I think this correctly describes this relationship—this is based on the idea that m, which is the slope of the line passing through the point of no change in Figures 1C– 1E (i.e. (fU(1), f(1))) and an arbitrary positive fold change (i.e. (fU(x), f(x)), should be equal to the slope of the analogous line for all other positive fold changes. There might be a more elegant way to formulate this, though

3. As mentioned, I don’t think you need to make these changes—the current descriptions (using visual evaluation methods) are sufficient. However, being more exact here would help future researchers evaluate their transformations on the same criteria with less work, 

and it would open the possibility for objectively quantifying how well a transform adheres to these criteria.

1. (It might also make it easier to adapt proportionality, symmetry, and dynamic range to color-based encodings like Figure 7—if you’re only dealing with numbers rather than plots, it should be easier to apply these properties to color gradients.)

4. If you choose to not make these changes (which is fine), I propose just adding a short note somewhere in the introduction (maybe alongside the proportionality and symmetry definitions) mentioning that the visual evaluation methods are slightly informal ways of testing these properties.

>> These are good points, but we wish to save this for future research. We added text to 114: “Additionally, each of these visualization properties could be formally defined by mathematical relationships between fold changes and transform outputs, but a more rigorous exploration of these properties is reserved for future research.”

3. Explaining “Dynamic Range”: Similarly, the new information about dynamic range (including Figure 1B) helps explain things a lot. Thank you!

1. It took me a while to understand the new changes, but I think I understand the point being made in the “Dynamic Range” section of the introduction now. To help other readers see your intent clearly, I have two suggestions:

1. Maybe explain more directly (in the “Dynamic Range” section of the introduction?) that, when visualizing data spanning many orders of magnitude, linear scales make it harder to distinguish small values—i.e. that these scales are dwarfed by “outliers.” This point is already made in the “Dynamic Range” section of the introduction, but (if this seems reasonable) I recommend going into more detail in the text about why exactly we see the effect in Figure 1B.

2. The sentences in the introduction explaining why linear and MAD plots have only medium dynamic range should be expanded in order to clarify this. Maybe refer to Figure 1B and/or Figure 8 from lines 117–118 and line 136? When I first read line 136, I was confused as to why MAD plots had only medium dynamic range; seeing Figure 8 helped everything click.

>> Added this sentence to line 98: “When data spans multiple magnitudes on a linear scale, large outlying data values overwhelm the axis spatial encoding, often leaving insufficient space to distinguish differences between small values (e.g. the crowding between small fold changes on the linear axis in Figure 1B).” And on line 129: “Linear plots of fold change have medium dynamic range because they use a linear axis tick label mapping (see illustrative example in Figure 1B, and Figure 8 for an example using real data).”

Minor points 

1. Abstract, line 35: This is an extremely minor point, but you use the term “MAD-FC plot” here (despite only defining the term “MAD plot” later in the paper). There is thus a very slight inconsistency.

>> Changed to “MAD plot”.

2. You could probably leave this as is (I do not think it is essential to fix), but maybe this sentence could be rewritten to be consistent with the rest of the paper—perhaps something like “We argue that MAD-FC transformed fold changes may yield more useful visualizations than log or linear transformed fold changes […]”? But the decision is the author’s call—whatever you think would be best.

>> That is fair point, we would prefer to leave as is because we added text to differentiate the raw MAD-FC transformed values versus the MAD plot. We want to de-emphasize using just the raw MAD-FC transformed values/ raw MAD plot.

3. Introduction, line 44–45: “Typically, scientists visualize fold change with a plot using a log or a linear transform, the latter of which presents raw fold change values (2).”

1. “with a plot” could probably be removed from this sentence, since I believe it goes without saying (very minor, though, so feel free to disregard).

>> Removed as recommended.

2. The term “linear transform” might confuse some readers—if it wouldn’t be too much trouble, I would recommend just removing the four uses of “linear transform” throughout the paper (since when you visualize the raw fold changes you’re not really transforming the data at all).

1. As a less dramatic alternative, maybe you could add a “simply” to the end of this sentence, i.e. “simply presents raw fold change values”, or otherwise rephrase it a bit to be extra clear that your use of “linear transform” just amounts to visualizing the raw fold changes.

>> Great point, we went with the alternative option.

3. A slightly more important issue: to my understanding, the reference to (2) here (Love et al. 2014, the DESeq2 paper) supports the first claim in this sentence (that people typically visualize log-transformed fold changes), but that paper doesn’t provide any examples of visualizing raw / linear-transformed fold changes. Unless I am missing something, I recommend moving the reference to (2) earlier in this sentence to be after you mention log fold changes, just to be clearer that (2) is only a reference regarding log fold changes.

1. If you know of any examples of papers that visualize raw fold changes, you could cite those here after citing Love et al. 2014—I think reference (5) (O’Shea et al. 2017, the source of the data in Figure 4D–F) should be sufficient. (Before I realized that (5) would be a good example of a paper that visualizes raw fold-changes, I dug up another example—Figure 4 of https://onlinelibrary.wiley.com/doi/full/10.1111/eva.13142.)

>> Moved reference (2) to earlier in sentence, added your suggested reference.

3. Introduction, line 45: minor phrasing issue—I suggest adjusting “Both transforms have a unique set of properties” to something that indicates that these transforms (or scales, see above point) do not have the exact same set of properties (which seems to be what the current sentence implies, although it’s clear to me what you mean).

1. I think the easiest way to fix this would just be adjusting the sentence’s start to say that “Each transform has” rather than “Both transforms have,” if this seems reasonable.

>> Great point, changed as recommended.

4. Introduction, line 56: The changes in this section are great; thank you! The only tiny comment I have is that you may want to adjust this line to say “With this encoding, [the raw] fold changes of […]”, rather than just “With this encoding, fold changes of […]”. This will make it unambiguously clear to the reader that 2, 1/2, 3, and 1/3 are raw / linear fold changes, rather than log- or MAD-FC-transformed fold changes.

1. This could also apply to line 58, I believe.

>> Added “raw” to both instances to clarify as recommended.

5. Introduction, line 62: minor grammar issue—I think you mean to say “values” and “locations” to match the plural tense of “visualized datapoints”. I bring this up because the use of the singular-tense words “value” / “location” could confuse readers.

>> Changed as recommended.

6. Introduction, line 66: the term “raw logarithmic axis tick labels” has not yet been defined, and the meaning is not immediately obvious without reading other parts of the paper (e.g. Figure 1D). I suggest either (1) briefly defining this term, (2) rewriting this sentence in a clearer way, or (3) referring to one of your figures to explain this term.

1. Option (3) might be the best; maybe you could say something like “[…] raw logarithmic axis tick labels (e.g. rightmost y-axis labels of Figure 1D) […]”?

2. This also applies to line 107 of the introduction.

>> Added reference to figure for both instances.

7. Introduction, line 73: Minor grammatical error (“magnitude” → “magnitudes”), I think.

>> Changed as recommended.

8. Introduction, line 74: I assume “transformed units” here refers to the linear / log / MAD-FC transformed fold changes; this is fine as is, but if feasible you may want to make it a bit clearer (since the term “transformed units” has not been defined yet and is a bit vague).

1. Maybe something like “[…] between the transformed fold changes and corresponding fold change units […]”? This could still be made clearer, but I think it would help.

>> Changed wording as recommended.

9. Introduction, line 78: Minor, but I’m not sure “trend line” is the correct term here—I think you could just say “line”, unless I’m misunderstanding something.

>> Removed “trend”.

10. Introduction, lines 98–99: “[…] a linear scale can typically capture 8 orders of magnitude on a log2 scale, or about 2.5 orders of magnitude on a log10 scale.”

1. The notion of “a linear scale … on a log scale” seems confusing to me. I think your intent is to say that a typically-sized plot can span about 2^8 or 10^2.5 units? I think saying “units” or (like in the abstract) “log2 space” would help clarify this.

2. Also: it’s a small point, but these numbers (8 units in log2 space, 2.5 units in log10 space) are repeated a couple of times throughout the paper (in the abstract, here in the introduction, in the results near figure 8, and in the discussion).

1. This is fine as is, but I propose just giving these estimates once (probably h

---

## [Editor Report · Decision Letter 3]

16 May 2024

MAD-FC: a fold change visualization with readability, proportionality, and symmetry

PONE-D-23-12801R3

Dear Dr. Corliss,

We’re pleased to inform you that your manuscript has been judged scientifically suitable for publication and will be formally accepted for publication once it meets all outstanding technical requirements.

Kind regards,

Stephen R. Piccolo

Academic Editor

PLOS ONE

Additional Editor Comments (optional):

The comments from the reviewers have been addressed, and I congratulate you on a fine paper!

---

## [Editor Report · Acceptance letter]

21 May 2024

PONE-D-23-12801R3 

PLOS ONE

Dear Dr. Corliss, 

I'm pleased to inform you that your manuscript has been deemed suitable for publication in PLOS ONE. Congratulations! Your manuscript is now being handed over to our production team.

Kind regards, 

on behalf of

Dr. Stephen R. Piccolo 

Academic Editor

PLOS ONE